# Development of mRNA Lipid Nanoparticles: Targeting and Therapeutic Aspects

**DOI:** 10.3390/ijms251810166

**Published:** 2024-09-22

**Authors:** Yaping Liu, Yingying Huang, Guantao He, Chun Guo, Jinhua Dong, Linping Wu

**Affiliations:** 1College of Pharmaceutical Engineering, Shenyang Pharmaceutical University, Shenyang 110016, China; liu_yaping@gibh.ac.cn (Y.L.); huang_yingying@gibh.ac.cn (Y.H.); chunguo@syphu.edu.cn (C.G.); 2Center for Chemical Biology and Drug Discovery, Guangzhou Institute of Biomedicine and Health, Chinese Academy of Sciences, Guangzhou 510530, China; he_guantao@gibh.ac.cn; 3University of Chinese Academy of Sciences, Beijing 100049, China; 4Key Laboratory of Immune Response and Immunotherapy, Guangzhou Institutes of Biomedicine and Health, Chinese Academy of Sciences, Guangzhou 510530, China

**Keywords:** lipid nanoparticles, mRNA vaccine, infectious disease, cancer, genetic diseases

## Abstract

Lipid nanoparticles (LNPs) have emerged as leading non-viral carriers for messenger RNA (mRNA) delivery in clinical applications. Overcoming challenges in safe and effective mRNA delivery to target tissues and cells, along with controlling release from the delivery vehicle, remains pivotal in mRNA-based therapies. This review elucidates the structure of LNPs, the mechanism for mRNA delivery, and the targeted delivery of LNPs to various cells and tissues, including leukocytes, T-cells, dendritic cells, Kupffer cells, hepatic endothelial cells, and hepatic and extrahepatic tissues. Here, we discuss the applications of mRNA–LNP vaccines for the prevention of infectious diseases and for the treatment of cancer and various genetic diseases. Although challenges remain in terms of delivery efficiency, specific tissue targeting, toxicity, and storage stability, mRNA–LNP technology holds extensive potential for the treatment of diseases.

## 1. Introduction

Lipid nanoparticles (LNPs), based on ionizable lipids, are the most commonly used messenger RNA (mRNA) delivery vectors and play a crucial role in the research and clinical translation of mRNA therapy. By encapsulating mRNA within a lipid core, LNPs safeguard against mRNA degradation, enhance cellular uptake, and facilitate cytoplasmic release upon cellular entry [1]. Notably, LNPs boast an excellent drug-loading capacity, high encapsulation efficiency, sustained release, high stability, low toxicity, and enhanced efficacy. Currently, LNPs are the best choice for delivering nucleic acids. To date, three nucleic acid medicines based on LNPs have been approved. Patisiran, based on DLin-MC3-DMA lipids, is an RNA interference therapeutic drug used for treating hereditary transthyretin-mediated amyloidosis (hATTR amyloidosis) [2]. BNT162b2, which is based on ALC-0315 lipids, and mRNA-1273, which is based on SM-102 lipids, are mRNA vaccines used to prevent COVID-19 [3,4]. These applications demonstrate the success of LNP delivery systems for gene drugs and vaccines.

The momentum of preclinical research and clinical trials for mRNA–LNP drugs is on the rise. However, the current focus predominantly on liver delivery results in a prevalence of liver-targeted delivery among LNPs. DLin-MC3-DMA (MC3) LNPs are the most representative vector for liver-targeted delivery. Both SM-102 and ALC-0315 are MC3 LNPs that primarily deliver mRNA to the liver after intravenous injection [5]. However, LNPs for extrahepatic-targeted mRNA delivery are still in the early stages of development. To fully utilize the advantages of mRNA technology, such as its low cost, high efficiency, and short developmental cycles, the development of an extrahepatic-targeted delivery strategy for mRNA–LNPs is of great significance. Current research is focused on enhancing the accuracy of mRNA–LNP delivery to specific cells, tissues, or organs to improve the therapeutic effect and minimize potential side-effects. In this review, we summarize the latest strategies for the targeted delivery of mRNA–LNPs to cells and tissues, including tissue-specific or cell-specific LNPs, following different injection modes. The review also highlights the preclinical and clinical research progress of mRNA–LNPs in the treatment of cancer, infectious diseases, and hereditary diseases. Finally, the current challenges and potential future research directions are briefly reviewed.

## 2. General Characteristics of LNPs

### 2.1. Composition and Structure of LNPs

In general, LNPs are composed of ionizable cationic lipids (ICLs), helper lipids, cholesterol, and polyethylene glycol (PEG) lipids (Figure 1a). ICLs are key factors that determine mRNA delivery and transfection efficiency [6]. Their acid dissociation constant (pKa) determines the ionization behavior and surface charge of LNPs, further affecting their stability and toxicity. The pH sensitivity of ICLs enhances in vivo mRNA delivery by reducing interaction with anionic blood cell membranes, thereby enhancing LNP biocompatibility. In acidic endosomal environments, protonated ICLs interact with negatively charged endosomal vesicle membranes, facilitating endosomal membrane disruption and promoting escape [7]. Helper lipids, such as 1,2-dioleoyl-sn-glycero-3-phosphoethanolamine (DOPE) and 1,2-distearoyl-sn-glycero-3-phosphocholine (DSPC), envelope the lipid–mRNA complex, thereby ensuring LNP stability. They can regulate the fluidity of LNP bilayers and improve the delivery efficiency by enhancing the LNP fusion with biological membranes. Additionally, these helper lipids may contribute to the escape of LNPs from the endosomes [8,9]. Cholesterol, a class of steroid compounds, possesses a distinctive structure that distinguishes it from other components of LNPs, regulating membrane fluidity by filling gaps between phospholipids. When bound to phospholipids at a low phase transition temperature, it can decrease membrane fluidity and increase the thickness of the bilayer membrane. When bound to lipids with a high phase transition temperature, it improves membrane fluidity and narrows the bilayer membrane. This property shields the LNPs from serum proteins, thereby enhancing their stability in biological environments [9,10]. PEG–lipids typically contain hydrophilic PEG chains and two flexible hydrophobic alkyl tails. PEG–lipids improve particle stability, reduce particle binding to plasma proteins in vivo, and prolong the systemic circulation time [11]. Additionally, PEG–lipids protect the LNP shell, inhibit LNP aggregation, and extend the formulation half-life. This influences LNP biodistribution in vivo and shields them from nonspecific endocytosis by immune cells (Table 1).

The core–shell structure of LNPs constitutes an efficient drug delivery platform capable of protecting and delivering various cargoes, including small-molecule drugs, mRNA, proteins, or peptides. The core is formed by the combination of these loads and lipid molecules through electrostatic or hydrophobic interactions. The shell is composed of lipids that encapsulate the core and exhibits good biocompatibility, which can effectively protect the core and enhance its stability. Functionalized auxiliary components are often modified on the surface of the shell to improve the cellular uptake efficiency and targeting of the nanoparticles. For example, PEG–lipids can be modified on the surface of the shell to form a hydrophilic protective layer, reducing recognition and clearance by the immune system and thereby prolonging the circulation time of LNPs in vivo. In addition, the targeting of LNPs to specific cells can be further enhanced by introducing specific targeting ligands (such as antibodies, peptides, and glycans). The design flexibility of LNPs makes them important tools in drug delivery and gene therapy. The pharmacokinetics, cellular uptake efficiency, and therapeutic efficacy of LNPs can be optimized by adjusting the composition of the core and shell layers. Several self-assembled LNP structures have been described in the literature, including multilamellar vesicles (Figure 1b) [12], nanostructured core particles (Figure 1c) [13], and homogeneous core–shell structures (Figure 1d) [14].

**Figure 1 ijms-25-10166-f001:**
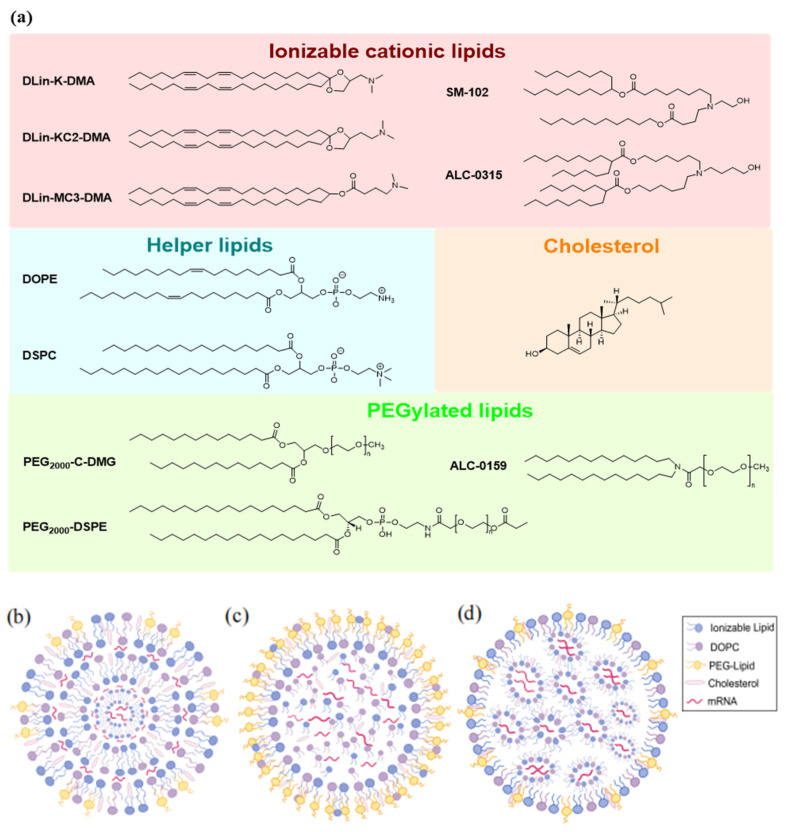
Composition and models of mRNA lipid nanoparticles: (**a**) Chemical structure of representative lipids used for delivery of therapeutic nucleic acids. (**b**) Multilamellar vesicle (onion-like). (**c**) Particle with a nanostructured core. (**d**) Homogeneous core–shell structure. (**b**–**d**) are reprinted with permission from [15]. Copyright 2023 Elsevier Inc.

**Table 1 ijms-25-10166-t001:** Typical composition and function of lipids in LNPs for mRNA delivery.

Component	Examples	Proportion (%)	Function	Modifications and Related Functions	Refs.
Ionizable lipid	DLin-MC3-DMA, ALC-0315, SM-102	30–50	Encapsulates payload mRNA, improves transfection efficacy, and enables differential delivery	Introduction of multiple tertiary amine nitrogen atoms: higher delivery efficiency and safety	[1,8,16,17,18,19,20,21,22]
Phospholipid	DOPE, DSPC, DOPG	10–20	Helper lipids considered to improve the structural stability of formulations	A 3-phosphate group modification: improves transfection efficiency; increases the distribution in vivo	[23,24]
Sterol	Cholesterol, cholesterol analogues	20–50	Structural stability	C-24 alkyl derivatives: regulate the integrity of the membrane; impact on delivery efficiency and biodistribution	[8,16,17]
PEG–lipid	PEG-DMG, PEG-DSPE	~1.5	Stealthiness: provides colloidal stability and enables evasion of the mononuclear phagocytic system	Covalent coupling modification: improving the half-life of LNPs	[1,18,19]

### 2.2. Toxicity of Lipids Used in LNP Formulations

Cationic lipids play a key role in LNPs by binding to nucleic acids through electrostatic interactions and facilitating the cellular uptake of LNPs through the cell membrane. However, these lipids may also induce cytotoxicity, especially at high concentrations. The cytotoxicity of cationic liposomes increases with increasing concentrations of cationic lipids [25,26,27]. For example, high concentrations of dimethyl dioctadecyl ammonium chloride (DODAC) in cationic liposomes correlate with enhanced toxicity to mammalian cells [28]. In addition, the toxicity of cationic lipids varies with their chemical properties. For example, lipids with quaternary ammonium head groups are more toxic than those with tertiary amine head groups [29]. Moreover, the composition and structure of liposomes significantly affect their cytotoxicity. For example, liposomes containing 50% DOPE induce approximately 80% mRNA silencing, whereas the degree of reduction in cell viability depends on the N/P ratio, which is the ratio of cationic lipids to phospholipids in the lipid, and the nature of the cationic lipids [30]. PEG–lipid couplers may also cause undesirable toxicity. Nanoparticles containing PEG–lipid couplers interact with immune cells, leading to the production of undesirable antibodies against PEGylated lipids. These toxic effects may affect the safety and efficacy of LNPs in vivo [31].

Therefore, to mitigate the cytotoxicity of LNPs, researchers need to carefully select and optimize the type and concentration of cationic lipids, as well as the composition and structure of liposomes. Such strategies include selecting less toxic cationic lipids, adjusting the ratio of cationic lipids to phospholipids, optimizing the composition and structure of liposomes, and avoiding the use of PEG–lipid couplings that may cause adverse immune responses [32]. These approaches may enhance the safety and therapeutic efficacy of LNPs.

### 2.3. Mechanism of mRNA–LNP Delivery

The mRNA–LNP delivery system undergoes essential processes to fulfill its functions in vivo. For example, mRNA–LNPs must be transported to different tissues and cells via the circulatory system [33]. Depending on the characteristics of the lipid molecules on the outer surface of the LNPs, they can interact with proteins on the cell membrane, thereby mediating the binding of mRNA–LNPs to specific cell membranes and initiating the endocytosis of LNPs. For instance, when the LNP surface is coupled with a CD5-specific antibody, the antibody binds to T-cells expressing the CD5 antigen, thereby mediating the endocytosis of mRNA–LNPs by T-cells [34]. This process endows LNPs with a targeting ability, which is of great significance in the development of targeted therapeutic methods for related diseases. Upon cellular internalization, LNPs form endosomes in the cytoplasm. Most LNPs are degraded by enzymes within endosomes; however, only a small fraction of mRNA–LNPs can complete the process of endosomal release, allowing exogenous mRNA molecules to enter the cytoplasm. With the assistance of cellular organelles such as ribosomes, the translation process is initiated, leading to the successful expression of exogenous proteins and the completion of the regulatory process of cellular functions. Most mRNA–LNP vaccines are internalized by cells after systemic administration. Subsequently, mRNA is released into the cytoplasm to guide the synthesis of antigenic proteins. These proteins are presented to T- and B-cells via major histocompatibility complex (MHC) molecules, activating the immune system to recognize and eliminate diseased cells while promoting tumor cell death (Figure 2).

## 3. Targeted Delivery of mRNA–LNPs

### 3.1. Cell-Targeted LNPs

Immune cells are important targets for nucleic acid delivery because they play a key role in many diseases, including cancer, inflammatory conditions, and autoimmune diseases. In addition to the reticuloendothelial system of the liver, immune cells are distributed throughout the body, particularly in the spleen. The delivery of therapeutic nucleic acids to leukocytes, such as macrophages, dendritic cells (DCs), and lymphocytes, provides an immune stimulation strategy to introduce genetic material with anti-inflammatory potential or to induce T-cell regulation. Targeting immune cells can directly affect the immune response, which is particularly important in treating diseases related to immune system dysfunction [36,37].

To enhance mRNA delivery efficiency and cell specificity, researchers have developed LNPs that target specific cells. These LNPs are designed for selective uptake by particular cells, facilitating efficient protein expression of the encapsulated mRNA within the target cells. Cell-specific uptake usually relies on ligand–receptor interactions in which the ligand binds to a specific receptor on the cell surface and promotes the internalization of LNPs. Surface modification is an effective strategy for enhancing the targeting ability of LNPs [38]. By attaching antibodies, peptides, carbohydrates, or other molecules to LNP surfaces, these molecules act as recognition signals, guiding LNPs to bind with target cells (Figure 3). For example, coupling antibodies against tumor cells with LNPs can target tumor cells more effectively in vivo, thereby improving the efficacy of the tumor treatment.

#### 3.1.1. Leukocyte-Targeted LNPs

Leukocytes represent a major therapeutic target because of their pivotal role in the immune system and their involvement in various diseases, including cancer, chronic inflammation, and autoimmune disorders [14]. These cells are ubiquitously distributed throughout the body, predominantly in organs such as the spleen and liver, as well as in the blood and lymphatic system.

Inflammatory bowel disease (IBD), a chronic autoimmune disease, remains incurable and causes chronic inflammation, severely impacting the quality of life of patients. To facilitate the cell-specific mRNA treatment of IBD, Veiga et al. [40] developed an antibody-modified LNP designed to target interleukin-10 (IL-10) mRNA to Ly6c^+^ inflammatory leukocytes. To elicit sustained IL-10 production, LNPs were initially formulated to encapsulate IL-10 mRNA. These LNPs were then incubated with Anchored Secondary scFv Enabling Targeting (ASSET) micelles at 4 °C for 48 h and subsequently with anti-Ly6c monoclonal antibodies (mAbs) for 30 min. The ASSET platform represents an innovative modular targeting system that facilitates the coupling of LNPs with targeting antibodies under mild conditions. As Ly6c^+^ leukocytes play a key role in inflammatory diseases, they are potential targets for the treatment of IBD [41]. Studies have demonstrated the inhibitory effect of the immunosuppressive cytokine IL-10 on IBD progression [42]. Various strategies for targeting leukocytes exist alongside this approach. For example, coupling a fusion protein associated with α4β7 integrins expressed on intestinal leukocytes to the surface of LNPs enables the silencing of interferon γ in the intestine and significantly improves the therapeutic efficacy in colitis [43]. Another approach is to design novel ionizable amino lipid libraries based on ethanolamine, hydrazine, or hydroxylamine ligands to efficiently deliver mRNA to leukocytes [44].

#### 3.1.2. DC-Targeted LNPs

DCs and antigen-presenting cells (APCs) play crucial roles in recognizing antigens and activating the immune response after the ingestion of foreign particles [45]. The internalization mechanism of DCs depends on phagocytosis or receptor-mediated endocytosis, which is significantly influenced by the particle size and surface modification. Given their ability to express antigen proteins directly, mRNA vaccines represent a promising strategy for activating the immune response, making DCs an ideal target for vaccine design [41]. Tang et al. [46] developed an mRNA vaccine modified with silicon-based neuraminic acid (SA) to effectively target DCs and escape endocytoses/lysosomes. SA-modified mRNA facilitated a twofold increase in the uptake of LNPs by DCs, with over 90% of the SA-modified LNPs escaping from early endosomes, avoiding lysosomal entry. This facilitated mRNA translation in ribosomes distributed in the cytoplasm and endoplasmic reticulum, significantly enhancing the transfection efficiency of mRNA–LNPs in DCs. Haase et al. [47] demonstrated that lipopeptide carriers are key components of highly efficient mRNA–LNPs. In particular, a series of lipophilic tertiary lipoamino fatty acid vectors with bundle-like topological structures (LAF4-Stp) and enhanced lipophilicity have exhibited excellent performance in effective mRNA encapsulation, cellular uptake, and endosomal escape. These novel LNPs exhibit superior transfection efficiency and faster transfection kinetics than conventional LNP formulations. Even with very low serum mRNA doses, high transfection levels were achieved. Upon intravenous injection of 3 µg of mRNA per mouse, in vivo mRNA expression was highly selective for DCs and macrophages, particularly in the spleen, where expression levels were significantly increased.

Elicitation of cellular responses by mRNA–LNP platforms relies on the activation of pattern recognition receptor (PRR) pathways and the engagement of DCs. Consequently, the incorporation of PRR ligands or mRNA encoding T-cell-polarizing cytokines into LNPs can induce the differentiation of specific T-cell subsets. An important strategy proposed to mitigate the inflammation caused by LNP components in mRNA–LNP vaccines is to target LNPs lacking cationic/ionizable lipids to DC subgroups that can initiate an antibody response in the absence of inflammatory factors [48].

#### 3.1.3. T-Cell-Targeted LNPs

LNP-based therapies targeting T-cells show great potential for the treatment of aggressive diseases, particularly cancer. Such therapies provide a novel strategy for treating various diseases by activating a patient’s immune system. In cancer treatment, mRNA–LNP technology can generate chimeric antigen receptor (CAR) T-cells that target tumors by delivering CAR mRNA, which can recognize and kill tumor cells [49]. Thatte et al. [50] developed a novel platform of ionizable LNPs designed for the efficient delivery of mRNA encoding the Foxp3 protein to CD4^+^ T-cells, enabling the engineering of Foxp3-T-cells in vitro. These Foxp3-T-cells temporarily exhibit an immunosuppressive phenotype and inhibit effector T-cell proliferation. This finding underscores the therapeutic potential of the LNP platform in modulating T-cell immunosuppression, which could significantly impact the advancement of autoimmune therapies. Tombácz et al. [51] studied a method for targeting human T-cells using Luc mRNA–LNPs coupled with an anti-CD4 antibody. Their findings revealed that anti-CD4/mRNA–LNPs resulted in stronger binding and dose-dependent luciferase expression in human CD4^+^ T-cells compared to control IgG-coupled LNPs. Yan et al. [52] showed that the subcutaneous administration of lead-induced ovalbumin (OVA) mRNA–LNPs can induce potent and durable CD8^+^ T-cell responses, substantially inhibiting the growth of aggressive B16-OVA tumors.

Cardiac fibrosis arises from the excessive production of the extracellular matrix by cardiac fibroblasts [53]. The efficacy of antifibrotic therapeutics in limiting fibrotic progression has not met expectations for the effective treatment of cardiac fibrosis [54]. The fibroblast activation protein CAR (FAP-CAR) is encoded by an mRNA [54], which is then encapsulated into LNPs modified with an anti-CD5 antibody to facilitate the specific targeting of CD5^+^ T-cells [55,56]. Rurik et al. [57] evaluated the therapeutic efficacy of reprogrammed CAR-T cells by injecting LNPs targeting CD5 into a mouse model of heart failure. The results showed that CD5-targeted LNPs effectively delivered therapeutic mRNA to lymphocytes in vivo, thereby forming transient anti-fibrotic CAR-T cells.

#### 3.1.4. Kupffer Cell- and Liver Endothelial Cell-Targeted LNPs

Kupffer cells, as macrophages in the liver, play a key role in immune monitoring and the inflammatory response of the liver. They are mainly responsible for removing particles and pathogens from the blood through phagocytosis [58]. To enhance the uptake of LNPs by Kupffer cells and their immunoregulation ability, research has explored various strategies, including increasing the size of LNPs and modifying the LNP surface with hydrophobic molecules [59]. Liver sinusoidal endothelial cells (LSECs) are located in the sinusoidal space of the liver, where they play important roles in blood filtration, metabolic regulation, antigen presentation, and lipid metabolism. LSECs have a unique structure and function, enabling them to play a crucial role in immune and metabolic processes in the liver [60]. Targeting LNPs to Kupffer cells and LSECs can lead to more effective drug or gene delivery, especially in the treatment of liver diseases. For example, in diseases that require liver-specific treatment, such as hepatitis, liver cirrhosis, and liver cancer, this targeted delivery strategy can improve the therapeutic outcomes and reduce unnecessary side-effects. Paunovska et al. [61] explored the use of different types of cholesterol to modify LNPs and reported that the structure of the cholesterol significantly affects the targeting ability of the LNPs. In addition, by using the Fast Identification Nanoparticle Delivery (FIND) system, they screened LNPs formulated with different oxidized cholesterol and observed that LNPs containing oxidized cholesterol can efficiently deliver mRNA to cells in the liver microenvironment, including Kupffer cells and liver endothelial cells. This contrasts with conventional LNPs, which primarily target hepatocytes, and suggests that the targeting of LNPs to specific cell types can be improved by specific chemical modifications. This improved targeting has significant implications for the treatment of liver diseases and the application of immunotherapy.

### 3.2. Tissue-Specific Targeting

By adjusting the ratio of lipids in LNPs, the targeting ability of LNPs to different organs can be tailored. Siegwart et al. [62] discovered that modifying the composition and components of LNPs allows for the selective targeting of organs. These passively targeted LNPs are termed selective organ targeting (SORT) nanoparticles, facilitating controlled nucleic acid delivery to specific tissues. Lung-targeting SORT LNPs feature a notably different protein corona from liver-targeting LNPs, with a reduced enrichment of ApoE, complement components, and immunoglobulins as well as increased enrichment of vitronectin (Vtn) among other proteins [63] (Figure 4). SORT LNPs contain SORT molecules and can accurately regulate delivery to the mouse liver, lungs, and spleen after intravenous injection [64] (Table 2). Siegwart et al. [65] conducted an exploration of the targeted mechanism of SORT LNPs. They speculated that different SORT molecules may cause LNPs to absorb different plasma proteins, thereby achieving tissue-specific mRNA delivery. Mass spectrometry analysis of the protein corona of different SORT LNPs revealed that the main proteins adsorbed by LNPs from the liver, lungs, and spleen were apolipoprotein E (apoE), vitronectin, and β-2-glycoprotein I, respectively. This suggests the potential to elucidate the targeting mechanism in extrahepatic tissues. Flux screening represents another method for developing tissue-specific LNPs. Dahlman et al. [66] developed a research strategy called the Joint Rapid DNA Analysis of Nanoparticles, utilizing DNA barcodes and sequencing techniques. This strategy enables the high-throughput screening of LNP formulations by encapsulating unique DNA sequences. Each LNP formulation is associated with a unique DNA barcode sequence. Using flow cytometry and next-generation sequencing techniques, hundreds of LNP formulations can be simultaneously analyzed to identify the relationship between LNPs and tissue- or cell-specific delivery [67].

#### 3.2.1. Liver-Targeted LNPs

The liver is an important organ that participates in digestion, excretion, detoxification, and immunity. Liver diseases are clinically common and frequently occurring. Some liver diseases, such as viral hepatitis, cirrhosis, and liver cancer, greatly endanger human health [90,91]. While many drugs are clinically used to treat liver diseases, most have limited clinical use owing to their poor distribution in the liver, significant toxic side-effects on other organs, and instability within the body. Consequently, there is an ongoing need to explore effective treatments for liver disease. Liver-targeted LNPs can effectively deliver drugs to lesion sites in the liver, reduce systemic distribution, decrease the dosage and frequency of medications, improve the therapeutic index of drugs, and reduce adverse reactions.

The success of patisiran (Onpattro^®^) demonstrates the potential application of liver-targeted LNPs in clinical treatment. This drug achieves efficient therapeutic effects by utilizing the natural uptake of LNPs by the liver and the expression of specific receptors on the surface of liver cells. During patisiran treatment, LNPs bind to apoE, which in turn binds to low-density lipoprotein receptors (LDLRs). Given the high expression of LDLRs on the surface of liver cells, this binding mechanism enables LNPs to effectively deliver small interfering RNA (siRNA) to the liver, thereby silencing the expression of abnormal proteins associated with hereditary thyroxine transporter amyloidosis. The therapeutic strategy of using LNPs for liver-targeted delivery has shown significant advantages in treating hepatocellular carcinoma (HCC), liver fibrosis, and other liver diseases [92]. Xu et al. [60] explored the possibility of delivering epitope mRNA–LNPs by targeting hepatic sinusoidal endothelial cells (LSECs), aimed at inhibiting allergic reactions to peanut crude protein extract. To achieve this goal, they developed mannose-modified LNPs that encapsulated mRNA-expressing peanut epitopes designed to enter the liver via biodistribution and be taken up by LSECs, which in turn induced a tolerance effect that effectively inhibited allergic reactions to crude peanut allergen extracts. This tolerance effect was accompanied by the inhibition of the Th2 immune response, IgE antibody production, and degranulation of mast cells. Kim et al. [93] reported the development of engineered LNPs for the targeted delivery of mRNA to hepatocytes and LSECs. They evaluated the effects of the particle size and PEG–lipid content of LNPs on the specific delivery of mRNA in the liver. Targeted mRNA delivery to the LSECs was further explored by introducing active ligands. Interestingly, Saunders et al. [94] utilized the dimensions of fenestrations in the liver and administered nanoprimers to mice as a pretreatment. This strategy aimed to reduce LNP uptake, consequently enhancing LNP accumulation in cell types other than Kupffer cells and LSECs.

Liver-targeted LNPs show potential for the treatment of hereditary diseases such as hemophilia and acute intermittent porphyria [95]. Studies have identified antithrombin (AT) as a negative regulatory factor that can be targeted to treat hemophilia A and B [96,97]. Restoration of the coagulation system balance has been achieved using the RNA interference drug fitusiran to selectively inhibit AT [98]. However, it is not a long-acting medication and requires repeated injections [99]. To seek long-acting treatments for hemophilia, Han et al. [69] developed LNPs encapsulating Cas9 mRNA and mouse AT-targeted single-guide RNA (sgRNA) for AT gene editing and inhibition. They evaluated therapeutic efficacy in hemophilia A and B mouse models by assessing thrombin generation mediated by mAT gene editing. They demonstrated that delivering CRISPR-Cas9 in vivo using LNPs could facilitate AT gene editing, thereby achieving the sustainable treatment of hemophilia A and B. The organ-specific delivery potential of LNPs was confirmed via the intravenous administration of luciferase-containing LNPs, resulting in luciferase expression primarily in the mouse liver, with no detectable expression in other organs. Recently, Genevant collaborated with Novo Nordisk to develop in vivo gene editing therapies for the treatment of hemophilia A by combining proprietary LNPs with innovative mRNA-based megaTAL technology [100]. Mitchell et al. [101] developed a high-throughput, low-cost synthetic strategy for constructing degradable branching-like lipids (DB-lipidoids), which can significantly improve mRNA delivery efficiency by up to threefold. The DB-LNPs, based on this strategy, exhibit high efficiency in liver mRNA delivery. DB-LNPs were used to deliver Cas9 mRNA/sgRNA for editing the *TTR* gene and mRNA encoding human fibroblast growth factor 21 (FGF21) to the liver as a strategy to treat the genetic disease transthyretin amyloidosis, and obesity and fatty liver, respectively. The results indicated that DB-LNPs achieved an approximately fivefold *TTR* gene editing efficiency and therapeutic FGF21 protein expression compared to MC3-LNP. These results suggest that DB-LNPs have great potential as a novel LNP platform for mRNA-based gene editing therapies and protein supplementation therapies.

#### 3.2.2. Extrahepatic-Targeted LNPs

Unlike hepatic targets, there are no specific proteins, such as apoE, that mediate the targeting pathways for extrahepatic targets. Consequently, several approaches have been employed for nonhepatic targeting. In this context, SORT was investigated by Dillard et al., in which a fifth component (or SORT molecule), in addition to the conventional four-lipid composition of LNPs, facilitates extrahepatic targeting [65]. Di et al. [102] have extensively reviewed methods for the extrahepatic targeting of mRNA delivery, which align with the SORT approach. Their findings indicated that positively charged lipids and negatively charged LNPs preferentially accumulate in the lungs and spleen, respectively.

The lungs represent a common target for nucleic acid treatment because of their association with numerous diseases and relatively accessible surface. However, they pose unique challenges due to their intricate biological mechanisms, such as mucociliary clearance and alveolar macrophage engulfment, which clear inhaled particles. Xue et al. [103] synthesized 180 cationic degradable (CAD) lipids and screened them using high-throughput DNA barcode technology to identify those that could effectively deliver mRNA to the lungs. Screening results showed that LNP-CAD9 efficiently delivered mRNA to the lungs, providing a potential novel therapeutic strategy for the treatment of lung diseases. Wei et al. [77] optimized and improved the Lung SORT LNP delivery system, aiming to effectively deliver Cas9 mRNA, sgRNA, and donor single-stranded DNA (ssDNA) templates to lung basal cells, thereby achieving precise homologous directed repair-mediated gene correction in a cystic fibrosis (CF) model. The optimized Lung SORT LNP system efficiently delivered mRNA to basal lung cells in Ai9 mice. Furthermore, LNPs can bypass pulmonary barriers and reach the lung endothelium through the bloodstream by modifying surface charge or functionalization using peptides, antibodies, or small-molecule ligands [81,104]. Massaro et al. [78] demonstrated the potential of the LNP delivery of mRNA therapeutics to lungs undergoing fibrosis. The utilization of small-sized LNPs ensures the efficient encapsulation of mRNA, offering effective protection against degradation and enabling the sustained release of mRNA. Notably, in vitro experiments confirmed the robust mRNA transfection efficiency, particularly at higher doses and longer intervals post-administration.

The inherent properties, composition, pKa, size, and charge modulation of LNPs offer significant potential for tissue-specific targeting. Further enhancement of ligand-based modifications has been recognized as a valuable strategy, effectively evaluated throughout the development of non-viral delivery vectors [40,51,105]. Longze et al. [106] developed a novel spleen-selective mRNA vaccine that can deliver unmodified mRNA and Toll-like receptor agonists to the spleen following systemic administration. This vaccine was designed to generate a sufficient and lasting anti-tumor cellular immune response, thus exerting a powerful tumor immunotherapeutic effect. Wang et al. [107] developed non-cationic thiourea lipid nanoparticles (NC-TNPs) to encapsulate mRNA through strong hydrogen-bonding interactions between the thiourea moiety of the NC-TNPs and the phosphate moiety of mRNA, discarding traditional anionic electrostatic interactions and constructing a non-ionized mRNA delivery system. The NC-TNP preparation technique is simple, convenient, and repeatable, and results in negligible inflammation and cytotoxic side-effects. It achieves a high gene transfection efficiency and can target the spleen for immune induction and disease treatment. Gomi et al. [82] developed a novel type of LNP loaded with phosphatidylserine (PS) to promote the targeted delivery of mRNA to the spleen. Compared with traditional LNPs, PS-LNPs show higher efficiency in delivering mRNA to the spleen. Sub-organ analysis showed that the PS-LNPs were mainly absorbed by macrophages in the red pulp and marginal area of the spleen, providing a promising strategy for the clinical application of mRNA in the spleen. Lymphoid organs contain high concentrations of immune cells and play a pivotal role in adaptive immune responses [108]. Efforts to target immune cells within these organs have been central to the development of vaccines and immunotherapies because they can elicit robust immune activation [22,109,110]. Ensuring the effective delivery and retention of vaccine candidates in the lymphatic system is essential for inducing protective immune responses, particularly for preventing infectious diseases (e.g., new coronaviruses and influenza viruses) [111,112]. Ramishetti et al. [113] showed that siRNAs can be efficiently delivered to CD4^+^ T lymphocytes by coupling anti-CD4 mAbs to LNPs.

In addition, the targeted delivery of mRNA to organs such as the brain, bone, heart, and eyes is a burgeoning research area. Tylawsky et al. [114] developed a fucoidan nanoparticle targeting P-selectin, which can penetrate the blood–brain barrier and transport anti-tumor drugs to brain tumor tissue. In a mouse model, these LNPs exhibited enhanced cancer treatment efficacy and reduced adverse effects. Mitchell et al. [115] designed 13 unique mRNA–LNPs based on different ionizable lipids. Through strict screening, “LNP A4” was identified as efficiently delivering mRNA to the placenta while avoiding fetal entry. This discovery has the potential to treat pregnancy complications such as pre-eclampsia. They used bisphosphate (BP)-modified LNPs to achieve the specific targeting of the bone microenvironment. A series of novel LNPs were prepared by introducing BP head groups into lipid molecules. These BP lipids can combine with calcium ions in bones through chelation, allowing the LNPs remain in the bone microenvironment for an extended period. This design enables LNPs to effectively accumulate in the bones, thus improving the effectiveness of drugs or gene therapy in the treatment of bone diseases [83].

## 4. Progress in Therapeutic Aspects

### 4.1. Treatment of Cancer

Cancer is a significant threat to human health worldwide, with rising incidence rates and high mortality rates presenting ongoing challenges to therapy [116]. Despite decades of research into tumor mechanisms and treatments, traditional approaches such as surgical resection, radiotherapy, and chemotherapy persist in clinical cancer management. However, these methods are not universally effective, and the toxicity and side-effects associated with radiotherapy and chemotherapy underscore the need for improved therapeutic strategies. Advances in science and technology have introduced numerous novel tumor treatment approaches in recent years, including biological therapies, particularly mRNA-based therapies. LNPs have emerged as an effective means of delivering mRNA to tumor cells, taking advantage of the unique characteristics of solid tumors such as high permeability and retention effects. Table 3 summarizes preclinical research on mRNA–LNP vaccines for cancer [117].

In cancer therapy, antibodies play a dual role: they directly modulate cancer cells and interact with the immune system to elicit both innate and adaptive immune responses [118]. It is assumed that all steps of the cancer immunity cycle function harmoniously to generate an effective immune response (Figure 5) [119]. In patients with cancer, the cancer immunity cycle may be impaired, resulting in an ineffective sequence and consequent lack of anti-tumor immunity, allowing cancer development and progression [120]. Although antibody therapies have demonstrated promising clinical efficacy, they have certain limitations. Challenges such as the stability, complexity of large-scale production, and substantial production and treatment costs may hinder their widespread application [88]. A promising approach involves producing antibodies in vivo by delivering the encoded mRNA, which can result in the effective in vivo expression of the desired antibodies [80,121]. Through sequence design, mRNA can encode therapeutic antibodies. Antibodies produced by mRNA have longer half-lives and lower costs. Various types of antibodies, including monoclonal antibodies (mAbs), bispecific antibodies (bsAbs), and their derivatives, can be expressed using mRNA. Wu et al. [122] showed that mRNA prepared by in vitro transcription (IVT) and formulated into LNPs could efficiently express endogenous therapeutic antibodies in hepatocytes, including intact pembrolizumab (an immune checkpoint inhibitor). This finding offers an alternative approach to antibody therapy for cancer treatment and highlights avenues for further research to optimize mRNA translation and secretion efficiency. This study also lays the foundation for the development of mRNA-coding therapies based on mAbs. Wang et al. [70] developed a bispecific antibody, BisCCL2/5i, which binds to the chemokines CCL2 and CCL5, thereby inhibiting immune cell chemotaxis. Using an mRNA nanoplatform, BisCCL2/5i can significantly induce the polarization of tumor-associated macrophages to the anti-tumor M1 phenotype and alleviate immunosuppression in the tumor microenvironment. BisCCL2/5i combined with a PD-1 ligand inhibitor (PD-Li) resulted in long-term survival in mouse models of primary liver cancer, colorectal cancer, and pancreatic cancer with liver metastasis. This study provides an effective bispecific targeting strategy for the treatment of malignant liver tumors and is expected to extend PD-Li therapy to many types of human liver malignant tumors.

The mRNA vaccine is first delivered to APCs, where tumor-associated antigens or neoantigens are produced. Subsequently, these expressed antigens bind to the MHC on the surface of APCs and are presented to T lymphocytes, activating CD4^+^ and CD8^+^ T lymphocytes, which then proceed to eliminate tumor cells (Figure 6). Compared with conventional vaccines, mRNA vaccines can stimulate stronger interferon type I responses and effectively activate human CD8^+^ T-cells [124]. These immune responses play a key role in the eradication of tumors. In a prior study, Oberli et al. [12] developed a library of LNPs to deliver an mRNA vaccine encoding a model antigen OVA to induce a cytotoxic CD8^+^ T-cell response. The efficacy of this vaccine was tested using an invasive B16F10 melanoma model. They observed robust CD8^+^ T-cell activation after a single immunization. Bevers et al. [125] optimized an mRNA–LNP formulation to enhance the delivery efficiency of an mRNA vaccine encoding human papillomavirus type 16 antigen. In a homozygous mouse TC-1 tumor model, the optimal LNP composition triggered a strong CD8^+^ T-cell response. This response could be further enhanced by repeated administration, resulting in notable anti-tumor effects. In addition, type I interferons and phagocytes play key roles in the T-cell response. These findings indicate the feasibility of using mRNA–LNPs as cancer vaccines.

In many types of tumor cells, tumor suppressor genes such as *PTEN* and *TP53* often exhibit dysfunction. Restoring the functions of these genes and their expressed proteins by mRNA delivery can regulate the anti-tumor response of cells and induce apoptosis. Approximately 36% of patients with HCC and non-small-cell lung cancer (NSCLC) carry *TP53* mutations [126]. Therefore, the restoration of *TP53* function may be a potential strategy for cancer treatment. Kong et al. [71] used LNP polymers consisting of G0-C14, PDSA, DMPE-PEG, and DSPE-PEG mixed with the mRNA encoding TP53 for cancer therapy. They found that a TP53 inhibitor not only inhibited the growth of tumor cells by inducing apoptosis and cell cycle arrest, but also increased the sensitivity of tumor cells to everolimus (an mTOR inhibitor). Kamath et al. [127] reported a redox-responsive LNP loaded with TP53-coding mRNA, and studied the therapeutic effects of this LNP in TP53-deficient Hep3B HCC and H1299 NSCLC cells. Apoptosis was significantly increased in both in vitro cellular experiments and an in vivo animal model.

mRNA-4157 is a novel individualized mRNA-based cancer vaccine that encodes 34 patient-specific tumor neoantigens. Currently, its efficacy in the treatment of melanoma is being evaluated. According to recent clinical trial data, the combination of mRNA-4157 and pembrolizumab has shown remarkable efficacy in the treatment of high-risk melanoma. Weber et al. [128] evaluated whether mRNA-4157 in combination with PD-1 could improve recurrence-free and distant metastasis-free survival in patients with high-risk melanoma. The results indicated that combination therapy resulted in longer recurrence-free survival and a lower rate of recurrence or death than monotherapy. This combination is currently being assessed in phase III clinical trials.

The delivery efficiency and biosafety of mRNA can be further improved by modifying the chemical structures of LNP components. Zhang et al. [129] described a fluorinated modification of DSPE-PEG2000, termed FPD. Despite constituting only 1.5% of the lipid components in LNPs, FPD significantly improved the mRNA expression efficiency in B16F10 tumor cells and primary DCs by fivefold and nearly twofold, respectively. Moreover, a mechanistic study showed that FPD facilitates the internalization and endosome escape of LNPs. Shin et al. [130] developed a novel delivery platform based on polyethyleneimine-modified porous Si nanoparticles (PPSNs) for local immunotherapy in vivo. This platform can effectively carry cytokine mRNA and translate it locally, thereby activating the immune response in the tumor microenvironment and avoiding the toxicity of systemic administration. The direct injection of PPSNs with cytokine mRNA into tumors can lead to high-level protein expression in tumors and promote the death of immunogenic cancer cells.

These findings underscore the potential of IVT-mRNA–LNPs as a novel approach for protein delivery, offering an effective strategy for cancer therapy. Compared to vaccines that stimulate active immunity, passive-immunity mRNA–LNP vaccines can rapidly and directly induce immune responses. Upon administration, the antibodies produced from the mRNA provide immediate, specific, albeit transient, immunity. Therefore, these vaccines require continuous administration to effectively treat tumors. Despite extensive research on the application of mRNA–LNPs in tumor therapy, their clinical applications remain limited. Thus, mRNA–LNP vaccines have broad research prospects in tumor therapy.

### 4.2. Treatment of Infectious Diseases

The mRNA vaccines represent an innovative immunization strategy for combating infectious diseases, offering technical advantages such as high flexibility and rapid response. Because the mRNA production process does not rely on live or inactivated viruses, the scale of production can be rapidly expanded without sacrificing safety. In addition, mRNA vaccines can be designed to rapidly adapt to viral mutations that are critical for responding to pathogens with high genetic variability such as influenza viruses and SARS-CoV-2 (COVID-19) [131]. During the COVID-19 pandemic, mRNA vaccines, including Moderna’s mRNA-1273 and Pfizer/BioNTech’s BNT162b2, encapsulated by LNPs, demonstrated exceptional preventive efficacy and safety in clinical trials. Both vaccines received emergency-use authorization from the FDA by the end of 2020 and have since been widely used worldwide, playing a key role in controlling the COVID-19 epidemic [132]. Besides the COVID-19 vaccine, mRNA technology is also expected to be used to develop vaccines against other infectious diseases such as influenza, AIDS, rabies, and Zika virus. Progress in this field offers a new platform for future vaccine research and development, potentially transforming our approach to managing infectious diseases. Table 4 summarizes the preclinical research for mRNA–LNP vaccines against infectious diseases.

Vaccines can effectively induce CD4^+^ and CD8^+^ T-cell responses, which are crucial in the fight against acute SARS-CoV-2 infection and the development of long-term immunity [144]. Notably, mRNA vaccination elicits significant T-cell responses, particularly after the second dose. For instance, one study found that, although circulating CD4^+^ T-cells and neutralizing antibodies were weakly detected one week after BNT162b2 vaccination, a robust CD8^+^ T-cell response was elicited, further enhanced after the booster vaccination [145]. Additionally, memory spike-specific CD8^+^ T-cells exhibited lower CD8^+^ expression during the early post-vaccination period compared to the natural infection period, potentially influencing long-term maintenance characteristics. Research and development of mRNA–LNP vaccine candidates is currently in the preclinical and clinical stages, covering a wide range of viruses, including HIV, seasonal influenza, Zika virus, respiratory syncytial virus (RSV), and Epstein–Barr virus. These vaccine candidates elicit strong protective immune responses against various viruses [146]. For example, mRNA-based influenza vaccines can stimulate immune responses against both heterologous and homologous influenza viruses [147]. The research team of Norbert Pardi cooperated with Professor Drew Weissman, the pioneer of mRNA vaccines, to develop an mRNA–LNP vaccine against *Borrelia burgdorferi*, the pathogen of Lyme disease. This vaccine used LNPs to deliver mRNA expressing outer surface protein A (OspA) of *B. burgdorferi*. In preclinical animal experiments, mice produced strong antigen-specific antibodies and T-cell responses following a single inoculation of this mRNA–LNP vaccine targeting OspA, thus preventing *B. burgdorferi* infection. In addition, this vaccine caused a strong memory B-cell response that could be activated after a long time to help prevent subsequent *B. burgdorferi* infection [135].

Furthermore, mRNA technology extends beyond vaccine development and is promising for therapeutic applications. For example, Pardi et al. [134] demonstrated that the administration of mRNA encoding an anti-HIV antibody encapsulated in LNPs can be used as passive immunotherapy against HIV-1. The importance of this finding is that the delivery system and route of administration of mRNAs have significant impacts on their biodistribution and efficacy in vivo. This study demonstrates the potential of mRNA vaccines in the fight against HIV-1 and provides a new perspective for future vaccine research and treatment strategies. In preclinical or clinical trials, LNPs are primarily administered via intradermal, subcutaneous, or intramuscular injections, facilitating the targeting of nearby draining lymph nodes and enabling the presentation of antigens to T-cells and exposure to B-cells, thereby activating the immune response [76,148,149]. The two most widely used COVID-19 mRNA vaccines are primarily administered via intramuscular injection, a method that allows the vaccine to be rapidly absorbed into the bloodstream, thereby inducing a systemic immune response. Regarding tumor vaccines, in addition to the above three pathways, alternative methods such as colon administration, nasal drip, or intratumoral delivery can also enhance the targeting and efficacy of LNPs [150,151]. The selection of an appropriate route of administration is critical for accelerating the development of targeted mRNA–LNP vaccines. Different routes of administration may affect the distribution, stability, and induction of immune responses against mRNA–LNPs. For example, for certain diseases such as respiratory infections or certain types of cancer, a specific route of administration may be required to ensure that the drug can effectively reach the target tissue or cell [152]. Figure 7 provides a summary of the representative administration routes and their applications in specific diseases.

### 4.3. Treatment of Inherited Diseases

Hereditary diseases arise from complex and challenging pathologies, often involving pathological DNA sequences in various organelles containing the genome. Treatments for hereditary diseases can be divided into three main categories: exogenous gene introduction and replacement of damaged sequences, non-targeted exogenous supplementation of deficient genetic material, and gene editing techniques to directly rectify genetic mutations [153]. Currently, there are no effective treatments for many hereditary diseases. The advent of mRNA–LNP technology has provided innovative ideas for the treatment of these diseases. For genetic diseases caused by single-gene mutations, such as cystic fibrosis (CF) and thalassemia, mRNA–LNP technology offers the possibility of disease treatment by repairing or replacing abnormal genes [154]. This targeted approach offers fewer potential side-effects and is considered a valuable addition to traditional therapeutic strategies for hereditary diseases. For example, in the treatment of CF, an inherited lung disease, mRNA–LNP technology is used to introduce an mRNA sequence that can repair mutations in the cystic fibrosis transmembrane conductance regulator *(CFTR)* gene. This therapy has shown remarkable results in clinical trials, helping to improve lung function and quality of life in patients with CF. Table 5 summarizes the preclinical research on mRNA–LNP vaccines for genetic diseases.

Currently, clinical trials of protein replacement therapies using LNP-mRNA formulations are focused on inherited metabolic disorders, including argininosuccinate lyase deficiency (ASLD), propionic acidemia, and methylmalonic acidemia propionic acidemia. The common feature of these disorders is the absence of key enzymes, leading to the accumulation of specific metabolites, which, in turn, triggers clinical symptoms. Supplementation of the required enzymes with LNP-mRNA preparations retards the progression of the disease and may improve the symptoms and quality of life of patients [156,159,160]. Argininosuccinate lyase deficiency (ASLD) is the second most common urea cycle disorder, accounting for 15–20% of all urea production disorders. Under normal conditions, arginine breaks down arginine into alanine and glutamate. However, in patients with ASLD, because of the absence of ASL, argininosuccinate cannot be effectively decomposed, leading to the excessive accumulation of nitrogen in the blood in the form of ammonia, resulting in hyperammonemia [161]. This condition can cause neurotoxicity, affecting central nervous system function. Jalil et al. [162] have proposed a potential therapeutic strategy targeting the c.1153C>T variant of ASL using a CRISPR base editor delivered by LNPs. Unlike viral vectors, this method offers liver targeting and can effectively address the metabolic phenotype of ASLD, thereby improving patient health. Propionic acidemia, a rare inherited metabolic disorder caused by the accumulation of toxic metabolites owing to defects in propionyl-CoA carboxylase (PCC), poses a threat to the lives of infants. Early screening, diagnosis, and timely treatment are critical for saving the lives of affected infants and improving their prognoses. However, therapeutic drugs targeting the etiology of PCC have not yet been approved. Koeberl et al. [163] first attempted to use the mRNA expression of intracellular proteins as a protein replacement therapy in patients with rare diseases. Participants receiving various doses of mRNA-3927 demonstrated good safety and tolerability, with no observed dose-limiting toxicity. In addition, this therapy continuously reduced the levels of harmful metabolites related to propionate in the body, thereby significantly reducing the probability of metabolic compensatory events in children.

Previous studies have shown that conventional LNP-mediated mRNA delivery is limited to the retinal pigment epithelium (RPE) and Müller glial cells, leaving photoreceptor cells, crucial for visual phototransduction, untargeted due to their inability to penetrate the neural retina. Herrera et al. [164] successfully delivered mRNA to photoreceptor cells by modifying LNPs with peptide ligands. Peptides are composed of amino acid sequences of different lengths, which can be synthesized naturally or artificially. The structure and bioactivity of peptide chains can be tailored based on factors such as the charge density, hydrophobicity, hydrophilicity, structural conformation, and chemical modification, particularly to facilitate crossing biological barriers. Peptides found wide-ranging applications in medicine, including enhanced drug delivery, imaging agents, and nanoparticle drug targeting. Translating mouse model experiments into more clinically relevant non-human primate experiments, researchers observed significant expression of target proteins in photoreceptors, Müller glial cells, and RPE, the potential effectiveness of LNP–mRNA therapeutics in treating inherited blindness.

## 5. Summary and Outlook

mRNA–LNP therapy is an innovative nucleic acid therapy method that uses lipid nanoparticles as carriers to deliver mRNA into cells, enabling the regulation of specific genes. This approach is characterized by strong targeting, high delivery efficiency, and relatively good safety, making it a promising treatment for certain diseases. The successful application of mRNA vaccines during the COVID-19 pandemic significantly advanced the development of mRNA–LNP technology, offering new possibilities for treating a wide range of diseases, such as cancer, infectious diseases, and genetic diseases. Given that the physical properties of LNPs, such as particle size, morphology, and surface properties, are significantly affected by the structure and composition of lipids, the efficacy, safety, and drug distribution of LNPs can be improved by optimizing the lipid composition. In addition, the development of targeted mRNA–LNP delivery technologies for specific tissues and cells could further improve the therapeutic efficacy and safety of mRNA–LNPs, expand the application of mRNA technology, and promote advancement of the field.

Non-viral vectors such as LNPs can efficiently load mRNA, improve the transfection efficiency, and trigger an immune response, thereby increasing antibody titers. Biodegradable LNPs can be rapidly eliminated from plasma and tissues, improving their safety and tolerability. Cationic lipids play a key role in determining the delivery efficiency and transfection efficiency of mRNA–LNP preparations, and the toxicity of the vector is related to the structure of the cationic lipids. To improve mRNA delivery efficiency, lipid head groups and hydrophobic tails can be modulated to increase the cellular uptake and endosomal escape of LNPs. In addition, the modulation of the lipid structure enables the cell-specific and tissue-specific delivery of LNPs. As the expression of mRNA in cells is transient, repeated injections of LNPs loaded with mRNA are often required to maintain the therapeutic effects. However, repeated injections may elicit an immune response, such as the production of antibodies against PEG, which may reduce the efficacy of the drug. Therefore, the search for disease-specific mRNA or the development of strategies that can prolong mRNA expression are important for improving drug efficacy and reducing the frequency of injections.

Overall, mRNA–LNPs represent a powerful and versatile platform for vaccine and drug development and are expected to emerge as a cornerstone in future tumor vaccine and drug development. While screening novel ICLs offers chemical innovation, predicting their delivery efficiency and targeting remains uncertain, posing challenges. Flux screening of LNP formulations, although beneficial, has limitations such as a heavy workload and poor organ specificity. Moreover, while SORT LNPs provide a clear research direction, the potential safety risks associated with the addition of cationic and anionic lipids require further data. Although antibody-modified LNPs offer significant advantages in cellular targeted delivery, addressing their shortcomings, such as poor tissue specificity and complex preparation processes, is imperative. In addition, attention should be directed toward designing the mRNA molecules themselves. Furthermore, in the era of rapid advancements in big data and artificial intelligence, their application in the development of targeted mRNA–LNPs could significantly enhance the research efficiency and success rates.

In summary, mRNA–LNP technology holds significant scientific and economic value. By employing various LNP development strategies, coordinating mRNA sequence research, and integrating advanced artificial intelligence technologies, novel targeted technologies are anticipated to emerge. The authors of this review anticipate successful delivery to more critical organs/cells and addressing unmet clinical needs in mRNA therapeutics.

## Figures and Tables

**Figure 2 ijms-25-10166-f002:**
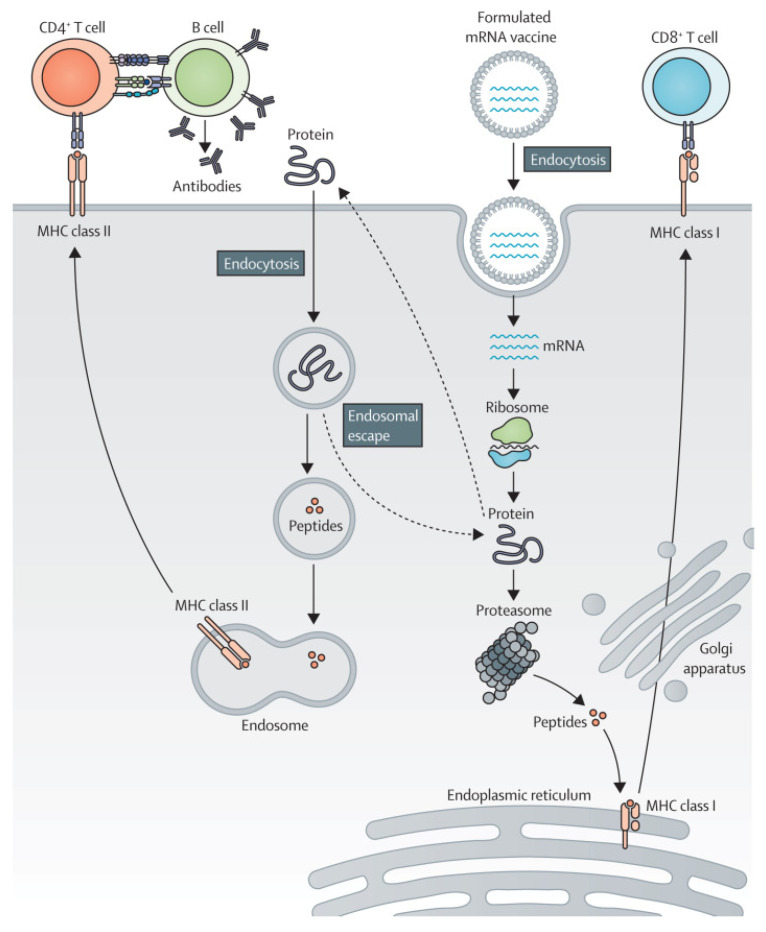
The mRNA–LNP vaccine immune response process. Reprinted with permission from [35]. Copyright 2022 Elsevier Inc.

**Figure 3 ijms-25-10166-f003:**
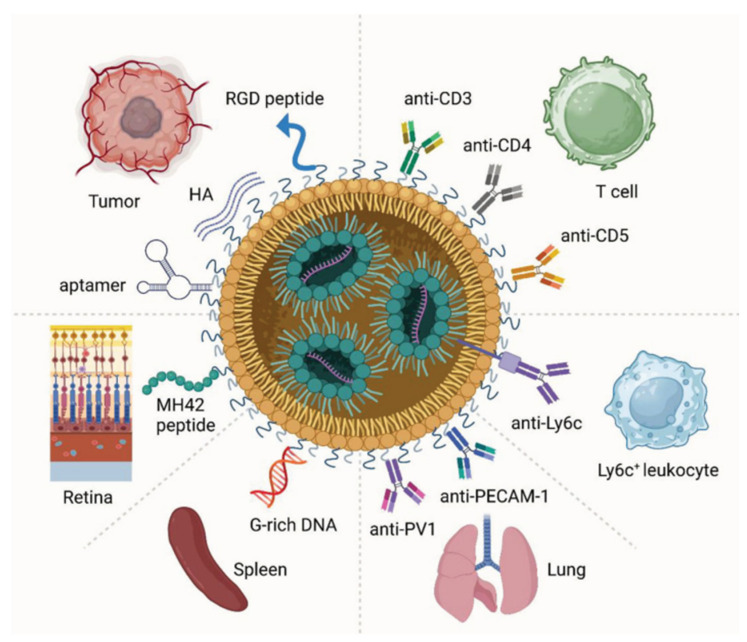
Representative surface modification strategies for LNP targeting. Reprinted with permission from [39]. Copyright 2023 Wiley.

**Figure 4 ijms-25-10166-f004:**
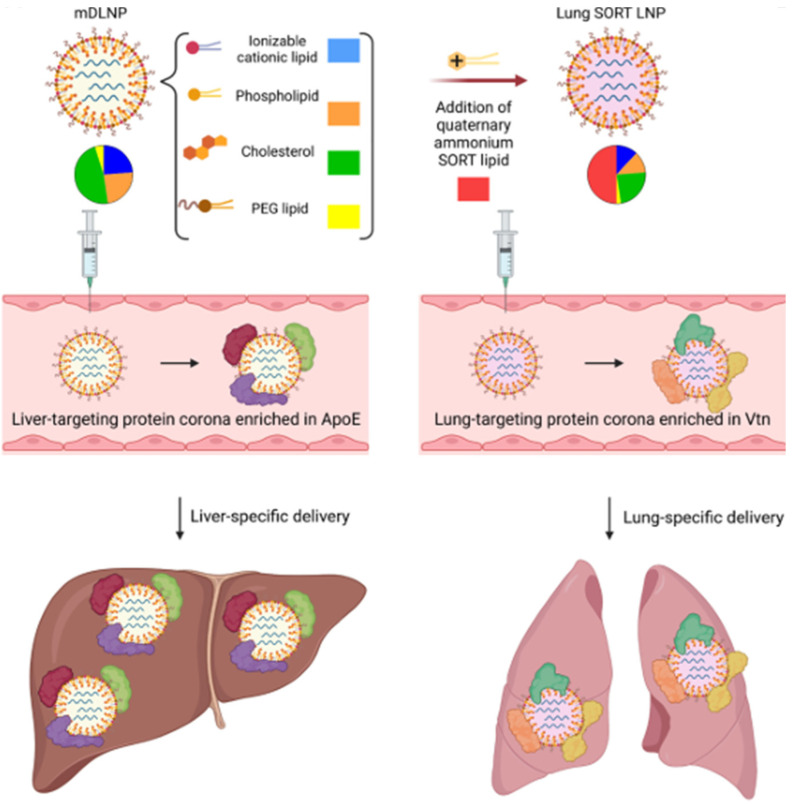
Selective organ targeting (SORT) technology achieves liver- and lung-specific mRNA delivery. Reprinted with permission from [63]. Copyright 2023 Elsevier Inc.

**Figure 5 ijms-25-10166-f005:**
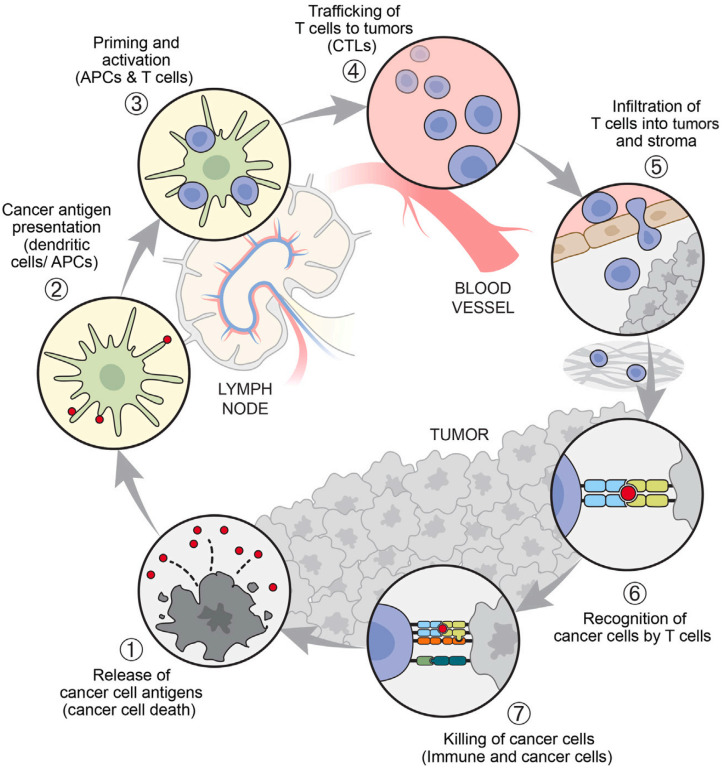
Key steps in the cancer immune cycle. Reprinted with permission from [123]. Copyright 2023 Elsevier Inc.

**Figure 6 ijms-25-10166-f006:**
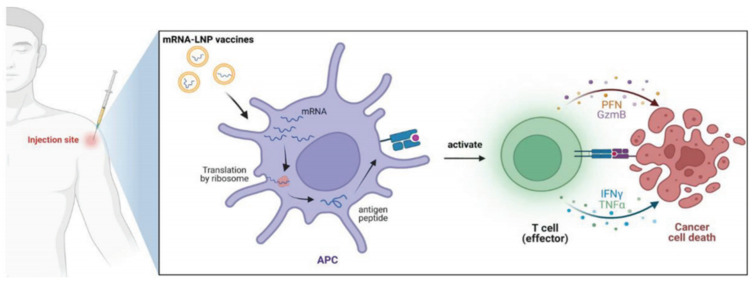
Schematic illustration of the mRNA–LNP cancer vaccines. Reprinted with permission from [39]. Copyright 2023 Wiley.

**Figure 7 ijms-25-10166-f007:**
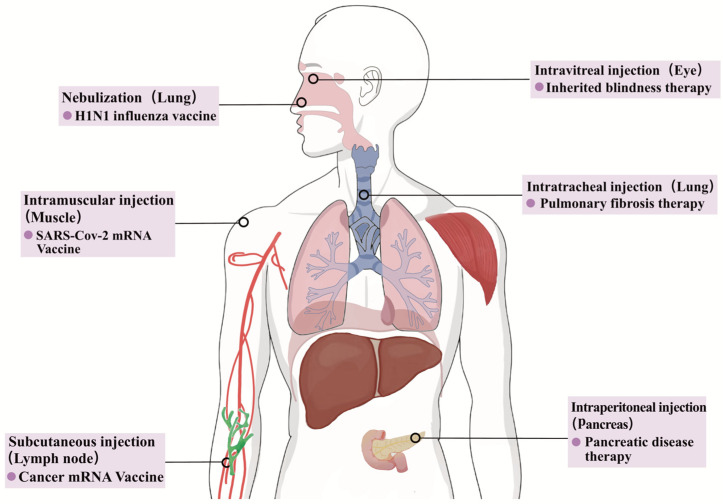
Representative administration routes and their applications of mRNA–LNP.

**Table 2 ijms-25-10166-t002:** Summary of different tissue-specific LNPs.

Disease	Target Organ	LNP Composition	mRNA	Administration	Ref.
Acute intermittent porphyria	Liver	2-(dinonylamino)-1-(4-(N-(2-(dinonylamino)ethyl)-N-nonylglycyl)piperazin-1-yl)ethan-1-one, DSPC, cholesterol, PEG-DMG	hPBGD mRNA	Intratumor injection	[68]
Hemophilia	Liver	246C10, DOPE, cholesterol, PEG–ceramide lipids	Cas9 mRNA and mouse AT-targeted sgRNA	Intratumor injection	[69]
Liver cancer	Liver	Dlin-MC3-DMA-based lipids, helper lipids, cholesterol, PEG–lipids	BisCCL2/5i mRNA	Intratumor injection	[70]
Liver cancer	Liver	G0-C14, PDSA, DSPE-PEG, DMPE-PEG	p53 mRNA	Intratumor injection	[71]
Liver cancer	Liver	5A2-SC8, DOPE, cholesterol, PEG–lipids	Cas9 mRNA, PD-L1 sgRNA, and FAK siRNA	Intratumor injection	[72]
SARS-CoV-2	Liver	MC3 lipids, cholesterol or β-sitosterol, DMG PEG2000, DSPC	hsACE2mRNA	Intratumor injection	[73]
Liver fibrosis and cirrhosis	Liver	PEG–lipids, ionizable lipids, structural lipids, cholesterol	HNF4A mRNA	Intratumor injection	[74]
Acute respiratory distress syndrome	Lung	DLin-MC3-DMA, DSPC, cholesterol, PEG2000-DMG	sPD-L1 mRNA	IV	[75]
H1N1 infection	Lung	Modified PEIcompound7C1, cholesterol, DMG-PEG2000, cationic lipid DOTAP	mRNA encoded with broadly neutralizing antibody targeting hemagglutinin	Inhalation	[76]
Cystic fibrosis	Lung	5A2-SC8, DOPE, cholesterol, DMG-PEG	Cas9 mRNA, sgRNA, and donor ssDNA templates	IV	[77]
Idiopathic pulmonary fibrosis	Lung	DLin-MC3-DMA, DPPC, cholesterol, DSPE-PEG	eGFP mRNA	intraperitoneal injection	[78]
LAM	Lung	306-N16B + 113-N16B, cholesterol, DOPE + DSPC, DMG-PEG2000	Cas9 mRNA, sgRNA, and Tsc2 mRNA	IV	[79]
Hematologic cancers and malignancies	Liver and spleen	IC8, DSPC, cholesterol, DMG-PEG	B7H3-CD3 mRNA	Intratumor Injection	[80]
NA	Spleen, lung, or liver	5A2-SC8 SORT lipids, DOTAB, cholesterol, DMG-PEG	Cas9 mRNA and sgRNA	Intratumor Injection	[81]
NA	Lymphoid tissues	ssPalm, phosphatidylcholine (PC), cholesterol, DMG-PEG2000	Cre-mRNA	IV	[82]
Skeletal diseases	Bone	Lipids with bisphosphonate head groups	BMP-2 mRNA	IV	[83]
Orthotopic glioblastoma	Tumor	DLin-MC3-DMA, DSPC, cholesterol, DMG-PEG, DSPE-PEG	Cas9 mRNA and sgPLK1	Intracerebral Injection	[84]
Melanoma	Tumor	DALs, DOPE, cholesterol, DMG-PEG	IL-12 mRNA	Hypodermic Injection	[85]
Melanoma	Tumor	C12-200 + CKK-12, DOPE + DSPC, cholesterol, C14-	TRP2/gp100 mRNA	Hypodermic Injection	[12]
Melanoma	Tumor	PEG1000 + C14-PEG350, arachidonic acid A2, DOPE, cholesterol, C14-PEG	OVA RNA	IV	[86]
Lymphoma	Tumor	Ionizable cationic lipids, phosphatidylcholine, cholesterol, PEG–lipids	Rituximab mRNA	IV	[87]
Breast cancer	Tumor	cKK-E12, DSPC, cholesterol, PEG2000-DMG	HER2 antibody mRNA	IV	[88]
Melanoma	T-cells	PL1, DOPE, cholesterol, DMGPEG2000	OX40 costimulatory receptor mRNA	IV	[84]
Cardiac fibrosis	T-cells	Anti-CD5 antibody, ionizable lipids, phosphatidylcholine, cholesterol, PEG–lipids	FAP-CAR mRNA	IV	[57]
Inflammatory bowel disease	Leukocyte	Anti-Ly6cmAbs, MC3, DSPC, cholesterol, DMG-PEG, DSPE-PEG	IL-10mRNA	IV	[40]
NA	Kupffer cells and liver endothelial cells	Oxidized cholesterol, cKK-E12, PEG–lipids, DOPE	Cre mRNA	IV	[61]
Peanut allergy	LSECs	MC3, DSPC, cholesterol, DSPE-PEG2000-mannose	Ara h2 mRNA	IV	[60]
NA	SECs/LSECs	MC3, DSPG, cholesterol, DMG-PEG2000 eGFP mRNA, mCherry	eGFP mRNA or mCherry	IV	[89]

IV: intravenous injection; hsACE2: human angiotensin-converting enzyme 2; HNF4A: human hepatocyte nuclear factor alpha; hPBGD: human porphobilinogen deaminase; LAM: lymphangioleiomyomatosis; NA: Not applicable; LSECs: Liver sinusoidal endothelial cells; SECs: sinusoidal endothelial cells.

**Table 3 ijms-25-10166-t003:** Preclinical research of mRNA–LNP vaccines for cancer treatment.

Disease	Vaccine	Lipid Nanoparticle Components	Molar Lipid Ratios	Target Antigen	Route	Ref.
Lymphoma	Rituximab mRNA-LNP	ALC-0315 lipids, phosphatidylcholine, cholesterol, PEG–lipids	50:10:38.5:1.5	Rituximab	ID	[87]
Melanoma	TRP2/gp100- mRNA-LNP	C12-200 + CKK-12, DOPE + DSPC, cholesterol, C14-PEG1000 + C14-PEG350, arachidonic acid	31.5:10:36:2.5:20	Tumor-associated antigens GP100 and TRP2	IH	[12]
Melanoma	IL-12-mRNA-LNP	DALs, DOPE, cholesterol, DMG-PEG	20:30:40:0.75	Interleukin-12	IH	[85]
Melanoma	A2-mOVA-LNP	A2, DOPE, cholesterol, C14-PEG	45:10:42.5:2.5	OVA	ID	[86]

ID: intradermal injection; IH: hypodermic injection; OVA: ovalbumin.

**Table 4 ijms-25-10166-t004:** Preclinical research for mRNA–LNP vaccines against infectious diseases.

Virus	Vaccine	Lipid Nanoparticle Components	Molar Lipid Ratios	Target Antigen	Route	Ref.
HIV-1	Env-mRNA-LNP	L319, DSPC, cholesterol, PEG-DMG	55:10:32.5:2.5	HIV-1 Env gp160	IM	[133]
HIV-1	VCR01- mRNA-LNP	L319, DSPC, cholesterol, PEG-DMG	50:10:38.5:1.5	Light- and heavy-chain broadly neutralizing anti-HIV-1 antibody VRC01	IV	[134]
Lyme	OspA-mRNA-LNP	ALC-0315 lipids, DSPC, cholesterol, and PEG-DAG	NR	Outer surface protein A (OspA)	Subcutaneous injection	[135]
Rabis	RABV-G-mRNA	Ionizable amino lipids, phospholipids, cholesterol, PEGylated lipids	NR	Glycoprotein of the Pasteur strain	ID/IM	[136]
Ebola	GP-mRNA-LNP	DMAP-BLP, DSPC, cholesterol and PEG–lipids	50:10:38.5:1.5	EBOV envelope GP	IM	[137]
CMV	PC mRNA-LNP	DMAP-BLP, DSPC, cholesterol, PEG-DSG	50:10:38.5:1.5	CMV glycoprotein gB and pentameric complex	IM	[138]
RSV	F-mRNA-LNP	Asymmetric ionizable amino lipids, DSPC, cholesterol, PEG2000-DMG	58:30:10:2	F protein of RSV and its related variants	IM	[139]
IBV	HA/NA/NP/M2- mRNA-LNP	ALC-0315 lipids, phosphatidylcholine, cholesterol, PEGylated lipids	NR	HAs from two lineages	ID	[140]
ZIKV	ZIKV prM-E mRNA-LNP	Ionizable amino lipids, phospholipids, cholesterol, PEGylated lipids	NR	prM and E glycoproteins of ZIKV strain Brazil SPH2015	IP	[141]
DENV-2	PrME, E80, NS1-mRNA-LNP	D-Lin-MC3-DMA, DSPC, cholesterol, PEGylated lipids	50:10:38.5:1.5	Structural proteins prME and E80 and nonstructural protein NS1 from DENV-2	IM	[142]
Powassan	POWV prM-E-mRNA-LNP	ALC-0315 lipids, structural lipids, sterol, PEGylated lipids	50:10:38.5:1.5	POWV prM and E proteins	IP	[143]

IM: intramuscular injection; ID: intradermal injection; IV: intravenous injection; NR: no report; IP: intraperitoneal injection; CMV: Cytomegalovirus; RSV: Respiratory Syncytial Virus; IBV: Infectious Bronchitis Virus; ZIKV: Zika Virus; DENV-2: Dengue virus.

**Table 5 ijms-25-10166-t005:** Preclinical research of mRNA–LNP vaccines for genetic disease treatment.

Disease	Vaccine	Lipid Nanoparticle Components	Molar Lipid Ratios	Target Antigen	Route	Ref.
Hemophilia A	hFVIII-mRNA-LNP	F8-N6, DSPC, cholesterol, PEGylated lipids	50:10:38.5:1.5	Antihemophilic globulin (Factor VIII)	IV	[155]
MMA	hMUT-mRNA-LNP	Ionizable lipids, DSPC, cholesterol, PEGylated lipids	50:10:38.5:1.5	Human methylmalonyl-coenzyme A mutase (hMUT)	IV	[156]
Cystic fibrosis	MRT5005	DMG-PEG_2000_, DOPE, ETI, imidazole cholesterol ester	NR	Fully functional CF transmembrane conductance regulator protein	Nebulzatin	[157]
Propionic acidemia	mRNA-3927	SM-102, DSPC, cholesterol, PEGylated lipids	NR	*α* and *β* subunits of propionyl-CoA carboxylase	IV	[158]

IV: intravenous injection; NR: no report; MMA: Methylmalonic acidemia.

## Data Availability

Data sharing not applicable to this article.

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
