# Peer review of "Development of mRNA Lipid Nanoparticles: Targeting and Therapeutic Aspects"

_ijms, 2024, doi:10.3390/ijms251810166_

Round 1

Reviewer 1 Report

Comments and Suggestions for Authors

The review discusses the use of mRNA lipid nanoparticles for treating infectious, malignant, and certain genetic diseases. It also explores the delivery of RNA therapeutics using lipid nanoparticles. It appears that this review may have been influenced by recent reviews, using similar titles and content. This lack of originality prevents it from standing out in the field. Also, I would like to highlight some key points from the review.

- The review should clarify its unique contribution compared to previous ones.

- In the introduction, the authors mentioned three nucleic acid medicines based on LNPs, but it's unclear which one is approved as an anticancer drug, and which is a vaccine. This should be specified.

- In Table 4, the authors labeled some papers as "NR" for "No Report" in the route of administration, but the route of administration is actually mentioned in those papers, so these should be revised.

- Additionally, the toxicity of lipid nanoparticles was not addressed.

- Some new reviews were missing from the references.

- Finally, the conclusion needs improvement to provide a more informative summary. 

Reviewer 2 Report

Comments and Suggestions for Authors

In the article, the authors provide a comprehensive review of the current state of mRNA lipid nanoparticles, focusing on their therapeutic applications and the specificity of their action on different targets. A general introduction to their characteristics and mechanisms of release is presented. The article discusses, in a general manner, what makes a specific type of LNP special for a particular disease or target and describes its possible mechanism of action, providing numerous detailed examples. Finally, the current state of these particles in the treatment of cancer, infectious diseases, and hereditary diseases is reviewed.

The article is well-written. Although there are numerous recent reviews on this type of therapeutic agent, some more general and others focused on specific details, the mention of some of them is missing. The main difference in this review is the exhaustive analysis of the examples, detailing the mode of action of each in relation to a specific disease. This compendium of information can serve as a valuable guide for professionals working in the field.

I believe the article is publishable in its current state after including a few references.

Some of the revisions, among others, I am referring to that should be included are:

Hou, X., Zaks, T., Langer, R. et al. Lipid nanoparticles for mRNA delivery. Nat Rev Mater 6, 1078–1094 (2021). https://doi.org/10.1038/s41578-021-00358-0

Recent Advances in Site-Specific Lipid Nanoparticles for mRNA Delivery, Xiao Xu and Tian Xia, ACS Nanoscience Au 2023 3 (3), 192-203; DOI: 10.1021/acsnanoscienceau.2c00062

Li X, Qi J, Wang J, Hu W, Zhou W, Wang Y, Li T. Nanoparticle technology for mRNA: Delivery strategy, clinical application and developmental landscape. Theranostics. 2024 Jan 1;14(2):738-760. doi: 10.7150/thno.84291. PMID: 38169577; PMCID: PMC10758055.

Round 2

Reviewer 1 Report

Comments and Suggestions for Authors

The authors have responded to all of the comments.